# Mitigating Traumatic Brain Injury: A Narrative Review of Supplementation and Dietary Protocols

**DOI:** 10.3390/nu16152430

**Published:** 2024-07-26

**Authors:** Federica Conti, Jackson J. McCue, Paul DiTuro, Andrew J. Galpin, Thomas R. Wood

**Affiliations:** 1School of Physics, University of Sydney, Sydney, NSW 2050, Australia; federica.conti@sydney.edu.au; 2School of Medicine, University of Washington, Seattle, WA 98195, USA; jjmccue@uw.edu; 3Department of Exercise Science, University of South Carolina, Columbia, SC 29208, USA; 4Center for Sport Performance, California State University, Fullerton, CA 92831, USA; agalpin@fullerton.edu; 5Department of Pediatrics, University of Washington, Seattle, WA 98195, USA; 6Institute for Human and Machine Cognition, Pensacola, FL 32502, USA

**Keywords:** traumatic brain injury, post-concussion syndrome, nutraceuticals, supplements

## Abstract

Traumatic brain injuries (TBIs) constitute a significant public health issue and a major source of disability and death in the United States and worldwide. TBIs are strongly associated with high morbidity and mortality rates, resulting in a host of negative health outcomes and long-term complications and placing a heavy financial burden on healthcare systems. One promising avenue for the prevention and treatment of brain injuries is the design of TBI-specific supplementation and dietary protocols centred around nutraceuticals and biochemical compounds whose mechanisms of action have been shown to interfere with, and potentially alleviate, some of the neurophysiological processes triggered by TBI. For example, evidence suggests that creatine monohydrate and omega-3 fatty acids (DHA and EPA) help decrease inflammation, reduce neural damage and maintain adequate energy supply to the brain following injury. Similarly, melatonin supplementation may improve some of the sleep disturbances often experienced post-TBI. The scope of this narrative review is to summarise the available literature on the neuroprotective effects of selected nutrients in the context of TBI-related outcomes and provide an evidence-based overview of supplementation and dietary protocols that may be considered in individuals affected by—or at high risk for—concussion and more severe head traumas. Prophylactic and/or therapeutic compounds under investigation include creatine monohydrate, omega-3 fatty acids, BCAAs, riboflavin, choline, magnesium, berry anthocyanins, Boswellia serrata, enzogenol, N-Acetylcysteine and melatonin. Results from this analysis are also placed in the context of assessing and addressing important health-related and physiological parameters in the peri-impact period such as premorbid nutrient and metabolic health status, blood glucose regulation and thermoregulation following injury, caffeine consumption and sleep behaviours. As clinical evidence in this research field is rapidly emerging, a comprehensive approach including appropriate nutritional interventions has the potential to mitigate some of the physical, neurological, and emotional damage inflicted by TBIs, promote timely and effective recovery, and inform policymakers in the development of prevention strategies.

## 1. Introduction

Traumatic brain injuries (TBIs) constitute a significant public health issue and a major source of disability and death in the United States and worldwide [1]. While precise incidence rates can be difficult to quantify due to the elusive nature and frequent underreporting of many head traumas, recent estimates clearly point to upward trends in the prevalence of TBIs across the globe. For example, according to the Centers for Disease Control and Prevention (CDC), in 2014 there were approximately 2.87 million TBI-related emergency department visits, hospitalisations, and deaths in the US alone, representing a 53% increase from 2006 [2]. Notably, TBI has become the signature wound of modern warfare with approximately 500,000 US Service members suffering a traumatic brain injury over the past 20 years [3].

In several specific age groups and recreational contexts, published TBI incidence figures are most likely inaccurate due to a multiplicity of confounding factors hindering timely and correct diagnosis of the injury [4]. For example, cases of TBI in older populations are often missed or misdiagnosed due to a significant overlap of symptoms with other age-related medical conditions like mild cognitive impairment and dementia [2]. Similarly, there is reason to believe that the occurrence of sport-related head traumas in young adults is largely underestimated given that concussions during training or sports events are not always reported or fully investigated [5]. Even in the face of significant under-reporting, known cases of TBI are associated with high morbidity and mortality rates [6]. As a result, TBIs represent a significant burden on healthcare systems, both in terms of direct costs (hospital admission, nursing services, testing procedures, medications, etc.), and with respect to productivity losses due to long-term disability and caregiving duties [7].

As defined by the CDC, a TBI results from a bump, blow, or jolt to the head, or a penetrating head injury, that disrupts the normal function of the brain. Despite a certain degree of individual variability in the symptomatology and neurological profile of the disease, pathological commonalities across all types of TBI comprise excitotoxicity, ionic disturbances, decreased cerebral blood flow, oedema, oxidative stress, inflammation, and damage to and death of brain cells that may occur during acute (i.e., within minutes) and subacute phases (i.e., within 24 h) [8], and potentially continuing for months or decades after the injury [9]. This combination of biological processes and protracted timeline, compounded by heterogeneous causes and presentation, makes TBI a complex and multifaceted disease affecting a constellation of organs and systems in the body and having significant physiological, psychological, and neurological repercussions on an individual’s health and well-being [10,11,12,13]. Since most cases of TBI are not accompanied by any change in physical appearance that could be easily detected, medical professionals often refer to TBI as the “invisible” injury. Taken together, the heterogeneity and invisibility of TBI represent a significant barrier to both timely diagnosis and effective treatment [14].

Among the many types of TBIs, mild TBIs are the most common [15] and involve a brief, if any, change in mental state or consciousness lasting no longer than 30 min [16]. Confusion and post-traumatic amnesia can also be experienced for a couple of hours to less than a day. Recreational- and sport-related concussions are typically categorised as mild TBI (mTBI) as they tend to fall on the lower side of the severity spectrum, often times being accompanied by no loss of consciousness [17]. Moderate TBIs are characterised by loss of consciousness and amnesia lasting between 30 min and 24 h. Common symptoms include headaches, confusion, dizziness, nausea, vomiting, slurred speech, drowsiness, and difficulty concentrating. Finally, severe TBIs cause loss of consciousness and amnesia lasting more than 24 h and, in some cases, more than 7 days. Disease profiles closely resemble those observed in moderate TBIs, often accompanied by symptoms such as memory and attention problems, decision-making difficulties and learning impairments [18,19]. Injury-related structural abnormalities may also emerge on brain imaging regardless of TBI severity [20]. For more details on TBI categorisation, readers are directed to comprehensive reviews and previous work [16,17].

Due to the extensive physical, cognitive, and behavioural impact of TBIs, and the frequent persistence of symptoms, concerted efforts are required to minimise their pathological consequences. While improvements in the in-hospital management of more severe or penetrating head traumas have been made, options for sports-related concussions are less clear [21,22]. Barriers to understanding the efficacy of intervention for both TBI and concussion include study design (small sample size), absence of standardised clinical management and outcome measures, and poor understanding of the underlying mechanisms. More knowledge is required to identify key imaging and molecular biomarkers of TBI to deepen our understanding of associated pathological mechanisms. In addition, additional prevention strategies and neuroprotective therapies could reduce the incidence of TBIs and improve patient functional outcomes.

To this purpose, one promising avenue that has increasingly drawn attention is the design of TBI-specific supplementation and dietary protocols. Emerging evidence from experimental trials and systematic reviews suggests that certain micronutrients and biological compounds may have beneficial effects both before and after the occurrence of TBI and concussion by targeting specific neuropathological mechanisms such as inflammation and oxidative stress [23,24,25]. Prophylactic and/or therapeutic compounds currently under clinical investigation include creatine monohydrate, omega-3 fatty acids (Docosahexaenoic acid, DHA, and Eicosapentaenoic acid, EPA), branched-chain amino acids (BCAAs), riboflavin, choline, magnesium, berry anthocyanins, Boswellia serrata, enzogenol, N-Acetylcysteine (NAC) and melatonin.

The scope of the present narrative review is to summarise the current literature on the neuro-protective effects of supplements in the context of TBI and downstream neurocognitive and other related symptoms and to provide an evidence-based overview of current supplementation and dietary protocols that may be considered in individuals affected by—or at high risk for—concussion and more severe head traumas. As such, emphasis will be largely placed on human studies. However, when human data on specific nutritional interventions are not available, we will discuss evidence from animal models to illustrate potential benefits and mechanisms of action for future studies to investigate.

While a number of investigational compounds may qualify as supplements relevant to TBI, we have chosen to focus on those that are readily available to the consumer. Importantly, though most of the relevant research has been conducted in TBIs of significant severity, we argue that clinical evidence favouring supplemental efficacy could be extended to milder brain injuries based on mechanistic plausibility, and because of the strong safety profile of the nutritional compounds under examination. In fact, the heterogeneous nature of the injury, especially within the category of mTBI, warrants careful consideration. “Mild” TBIs include everything from sport-related concussions where the athletes independently walk off the field to much more severe injuries that involve significant loss of consciousness (up to 30 min) and the need for hospitalisation. The narrative approach of the present review is therefore aimed at broadening the scope as much as possible while searching for positive asymmetry—interventions with low risk where there is potential for benefit despite the absence of large injury- or outcome-specific randomised controlled trials. As emerging research suggests, supplementation and dietary approaches have the potential to mitigate the physical, neurological, and emotional damage inflicted by TBIs across the spectrum of severity.

## 2. Methods

Relevant compounds discussed in the present review were identified through a search conducted using the MEDLINE (PubMed) and Cochrane online databases. We focused on recent systematic reviews and meta-analyses of relevant literature written in English, published in peer-reviewed journals and investigating the effect of dietary interventions on TBI-related health outcomes. Key search terms included nutrients, foods and biochemical compounds of interest (e.g., “melatonin”, “magnesium”, “kiwifruit”) or broader terms for post-TBI interventions (e.g., “supplements”, “nutraceuticals”) alongside “TBI”, “traumatic brain injury”, “concussion”, “post-concussion syndrome”. Asterisks (*) and Boolean operators were used to capture the derivatives of search terms. Potential studies were initially screened by reviewing titles and abstracts, followed by full-text review to confirm eligibility. Additional compounds were considered based on mentions in manuscripts found through the strategies above, and clinical and observational evidence from both animals and humans was included where necessary to provide additional information on efficacy or mechanisms.

## 3. A Brief Overview of Injury Mechanisms in TBI

The physiological consequences of TBIs involve complex pathological processes that develop over two different timescales: an initial, acute response accompanied by stretch-triggered neuronal excitotoxicity, and a delayed, secondary phase characterised by neuroinflammation and oxidative stress [26]. The more we understand how these neural processes arise and evolve over time, the more we can unravel their distinct physiological features and design specific dietary protocols to prevent and treat TBIs based on the mechanisms of action of each nutrient (see Table 1).

Excitotoxicity is a neurodegenerative mechanism triggered by the overactivation of glutamate receptors in the central nervous system [27,28]. Elevated glutamate concentrations lead to increased sodium and calcium influx through the plasma membrane [29], which initiates a cell degradation process involving the production of reactive oxygen species (ROS) [30], and the activation of several enzymes (e.g., proteases, lipases, nitric oxide synthase, and endonucleases) damaging cell structures, often to the point of cell death [31]. Elevated intracellular calcium further affects mitochondrial health and efficiency by increasing membrane permeability and inhibiting adenosine triphosphate (ATP) production [27], while the dysregulation of voltage-gated calcium channels in response to TBI-induced membrane depolarisation compromises cerebral vasoreactivity [32], thus contributing to neurovascular dysfunction [33,34]. In the attempt to reestablish membrane resting potentials, the brain then increases energy production via anaerobic glycolysis. However, the ATP supply coming from this cellular pathway is rather limited, and therefore quickly exhausted, leading to a net energy deficit [35]. Taken together, these post-injury adaptive processes disrupt cellular homeostasis, ultimately causing DNA fragmentation, apoptosis, necrosis, and cytoskeleton degradation [36,37].

The delayed phase of injury following head trauma is hallmarked by a pervasive inflammatory cascade [26] mediated by metabolic changes to the blood–brain barrier BBB, [38]. This physiological response facilitates the migration of peripheral immune cells into the brain [39] and the subsequent secretion of damaging cytokines [40]. Additionally, the activation of resident neuronal cells, including microglial cells, astrocytes, and complement proteins, releases neurotoxic substances including ROS, glutamate, and several cytokines [41], which have been shown to interfere with the natural healing process of the brain [40,42,43]. As part of the delayed phase, evidence for immune activation in the brain has been described several years after injury [9]. Importantly, it must be noted that the putative mechanisms of TBI have been elucidated in the preclinical setting and are difficult to confirm clinically. In addition, these mechanisms of injury occur on a continuum and would not necessarily be relevant to all injuries of all severities. For instance, axonal injury may include everything from complete shearing of the neuronal axon to stretch-triggered depolarisation depending on the severity of the impact or blast. Blast-related TBIs may also be more likely to involve vascular injury and disruption of the blood–brain barrier [44,45,46,47]. Therefore, while these mechanisms provide evidence to support certain interventions after TBI, more work is needed to better map specific mechanisms by mode and severity of injury to targeted interventions.

**Table 1 nutrients-16-02430-t001:** Mechanisms of actions and strength of evidence for nutrients of interest in the management of TBI.

Nutrient/ Biological Compound	Mechanisms of Action and Beneficial Effects on Brain Health	References	Study Type, Population, and TBI Severity	Strength of Evidence ^1^
Omega-3 fatty acids (DHA and EPA)	Decrease neuroinflammation. Attenuate NFL levels. Help preserve/increase brain and hippocampal volumes. Improve cerebrovascular responsiveness and cognitive function. Reduce the risk of AD.	Pottala, et al. [48]	CS, 1111 postmenopausal women (mean age 78.5 ± 3.6) from the Women’s Health Initiative Memory Study	3
		Oliver, et al. [49]	RCT, 81 US college football players	
		Howe, et al. [50]	RCT, 38 hypertensive adults (mean age 63.7 ± 2)	
		Patan, et al. [51]	RCT, 310 healthy adults (aged 25–49)	
		Sala-Vila, et al. [52]	Prospective observational study, 1490 older adults (mean age 73 ± 5.7) from the Framingham Offspring Study	
Creatine monohydrate	Helps maintain ATP levels in response to high energy demands post TBI. Supports cognitive health. Improves symptoms of psychiatric disorders. Preventative supplementation may reduce neural damage following brain injury.	Sakellaris, et al. [53]	Prospective, randomised, open-labelled pilot study, 39 children (aged 1–18) with moderate to severe TBI (GCS at admission between 3 and 9)	2
		Cook, et al. [54]	Blinded, repeated measure, placebo cross-over trial, 10 healthy rugby players (mean age 20 ± 0.5)	
		Borchio, et al. [55]	RCT, 20 healthy semi-professional mountain bikers (mean age 29.5 ± 9.3)	
BCAAs	Function as a nitrogen donor in glutamate and GABA production. Supplementation improves markers of cognition, decreases concussive symptoms, and ameliorates sleep disturbances.	Aquilani, et al. [56]	RCT, 40 patients (aged 14–64) with severe TBI and 20 age-matched controls	2
		Aquilani, et al. [57]	RCT, 41 patients (49.5 ± 21) with a posttraumatic vegetative or minimally conscious state	
		Elliott, et al. [58]	Prospective RCT, 26 veterans (mean age 49.2 ± 9.5) with chronic mild TBI (24.5 ± 8.1 years post injury)	
		Corwin, et al. [59]	RCT, 38 adolescents and young adults (aged 11–34) experiencing mild TBI within the 72 h preceding enrolment	
Riboflavin	Helps address the energy deficit and oxidative stress following TBI.	Kent, et al. [60]	RCT, 52 young adults (mean age 20 ± 0.5) experiencing a sport-related concussion within the 24 h preceding enrolment	3
Choline	Helps preserve the integrity of the BBB as well as cellular membranes. Attenuates brain oedema. Acts as a precursor to acetylcholine. Improves spatial and recognition memory performance. Strong safety profile.	Levin [61]	RCT, 14 young adults (mean age 22.5) with mild TBI	3
		Aniruddha, et al. [62]	RCT, 44 adults (mean age 36.5 ± 16.2) with mild TBI	
		Zafonte, et al. [63]	RCT, 1213 adults (aged 18–70) with TBI of all severities (mild: 66.5%, moderate/severe: 33.5%)	
Magnesium	Modulates excitotoxicity. Promotes functional and cognitive recovery. Improves behavioural deficits. Helps regulate intracellular calcium concentrations	Temkin, et al. [64]	RCT, 499 subjects (aged 14 and older) with moderate to severe TBI (GCS: 3–12)	3
		Standiford, et al. [65]	RCT, 17 adolescents (aged 12–18) with mild TBI (GCS > 13)	
Blueberry anthocyanins	Decrease neuroinflammation. Reduce oxidative stress. Improve cognitive function and memory performance. Regulate concentrations of BDNF and 4-HNE in brain tissue.	Krikorian, et al. [66]	Placebo-controlled trial, 16 older adults (mean age 78.2 ± 5.8) with memory decline	3
		Whyte, et al. [67]	Double-blind, cross-over trial, 21 healthy children (aged 7–10)	
		Boespflug, et al. [68]	RCT, 16 older adults (aged 68–92) with mild cognitive impairment	
		Whyte, et al. [69]	RCT, 112 older adults (aged 65–80)	
		Miller, et al. [70]	RCT, 37 older adults (aged 60–75)	
		Barfoot, et al. [71]	RCT, 54 healthy children (aged 7–10)	
		Krikorian, et al. [72]	RCT, 27 overweight adults (aged 50–65) with subjective cognitive decline	
Boswellia serrata	Reduces neuroinflammation via genetic and metabolic mechanisms. Downregulates the production of inflammatory cytokines. Improves global cognition.	Moein, et al. [73]	Double-blind randomised cross-over trial, 38 subjects (aged 15–65) with diffuse axonal injury (GCS ≤ 12)	2
		Baram, et al. [74]	RCT, 80 adults (aged 40–80), with focal ischemic signs persisting for >24 h	
		Meshkat, et al. [75]	RCT, 100 adults (mean age 36 ± 14) with TBI of all severities (mild: 17.15%, moderate: 47.45%, severe: 35.4%)	
Enzogenol	Supports cognitive function. Reduces mental fatigue and sleep disturbances.	Theadom, et al. [76]	RCT, 60 adults (aged 21–64) with mild TBI	2
		Walter, et al. [77]	RCT, 42 young adults (aged 18–24) with a history of sports-related concussion (six months to three years post-injury	
NAC	Decreases neuroinflammation. Improves neuropsychological and cognitive outcomes.	Hoffer, et al. [78]	RCT, 81 US active-duty service members (aged 18–43) with blast exposure mild TBI	3
Melatonin	Supports sleep quality (uncertain effect on daytime sleepiness).	Kemp, et al. [79]	Double-blind crossover pilot study, 7 men (aged 17–55) with post-TBI sleep disturbances (28.6% mild, 42.9% moderate, 28.6% severe)	2
		Grima, et al. [80]	Double-blind placebo-controlled crossover study, 33 adults (mean age 37 ± 11) with post-TBI sleep disturbances (6% mild, 9% moderate, 85% severe)	

Abbreviations: 4-HNE = 4-hydroxynonenal; AD = Alzheimer’s disease; ATP = Adenosine triphosphate; BBB: Blood-brain barrier; BDNF = Brain-derived neurotrophic factor; CS = cross-sectional study; GABA = Gamma-aminobutyric acid; GCS = Glasgow coma scale; NAC = N-acetyl cysteine; NFL = Neurofilament light; RCT = randomised controlled trial; TBI = Traumatic brain injury. ^1^ Strength of evidence are assigned following the Oxford Levels of Evidence criteria from the Oxford Center for Evidence-Based Medicine [81], based on research design, quality of relevant studies and applicability to patient care. A score of 1 represents the highest strength of evidence, as established by consistent results from systematic reviews of randomised controlled trials, down to levels 4 and 5, where supporting evidence proceeds only from case series, case-controlled studies, and mechanism-based reasoning. Levels may be graded down on the basis of imprecisions/limitations in study design, inconsistency or conflicting results between studies, or because the absolute effect size is very small. In the context of the present review, levels have also been graded down if relevant studies have only been conducted in animal species and/or non-TBI populations.

## 4. Supplementation and Potential Dietary Protocols for the Prevention and Treatment of TBI

In the following sections, we discuss the key micronutrients and biological compounds that have been suggested to improve structural and neurofunctional outcomes in the context of TBI. We illustrate the neurophysiological mechanisms underpinning their actions and interactions and present the findings of relevant research studies examining their safety and dose-dependent efficacy. Based on these results, we then summarise promising supplements and doses that may be considered by individuals at high risk of head trauma (Table 2). The majority of these nutrients are available from the diet, though the ability to derive them from food at clinically relevant doses will depend on both the composition of the diet and the specific compound. Some additional non-nutritive compounds of interest are also presented, based on evidence from small clinical TBI trials.

### 4.1. Positively Asymmetric Nutritive Compounds Derived or Available from Food

#### 4.1.1. Creatine Monohydrate

Creatine is a naturally occurring derivative of the amino acids methionine, glycine, and arginine. It is primarily stored in skeletal muscles but can also be found in the brain, testes, liver, and kidneys [82]. Creatine’s most notable function in the body is to maintain adequate levels of ATP in tissues that have high and fluctuating energy demands such as skeletal muscles and the brain [83]. By increasing concentrations of available phosphocreatine in cerebral tissue [53], creatine supplementation has been shown to support cognitive function and performance both in healthy individuals [84,85], and following head trauma [83], while also mitigating many of the downstream effects of TBI such as disordered sleep, altered cognition, and mood disturbances [82].

Those who suffer a TBI experience a cellular energy crisis placing high energy demands in an environment of reduced blood flow and hypoxia. Creatine can provide ATP to the brain during periods of low oxygen following head injury. However, if creatine stores are depleted, repeated blows to the head without restoring these stores can cause far greater damage, known as “second impact syndrome”. Patterns of multiple sub-concussive events are common in contact sports and military service. In a recent study, Ainsley Dean, et al. [82] showed a significant drop in creatine levels in the motor cortex and dorsolateral prefrontal cortex for football players over the course of a season.

Evidence from experimental models in rats demonstrates that pre-emptive supplementation with creatine may reduce damage following TBI by 36–50% in a “moderate” severity controlled cortical impact model that results in significant tissue loss [86]. Such neuroprotective effect has been suggested to stem from the creatine-induced preservation of mitochondrial function and membrane integrity, which in turn may help maintain adequate levels of intracellular calcium and ATP and regulate the release of ROS [86]. While preliminary, these results point to a potential dose–response mechanism whereby higher levels of creatine supplementation before the insult may help contain subsequent neural damage and lower critical biomarkers of TBI [87]. As creatine concentrations in the brain significantly decrease following head trauma, and neuronal uptake simultaneously slows down [82], benefits may be derived from the adoption of a preventive creatine supplementation protocol.

Notably, the use of creatine as a pharmacological tool for the treatment of TBI has also been evaluated at different developmental stages. For example, Sakellaris, et al. [53] conducted a randomised, open-labelled clinical trial to examine the effects of creatine supplementation in young children and adolescents (1–18 years old) affected by severe TBI (Glasgow Coma Scale, GCS 3–9). Starting at 4 h from the time of injury and for a duration of six months, a daily creatine dose of 0.4 g/kg of body weight was found to improve several TBI-related parameters and functional outcomes compared to placebo, such as the duration of post-traumatic amnesia, intubation and stay in intensive care units, the degree of disability and ultimate recovery from the insult, global cognition, as well as self-care, communication, locomotion, and sociability skills [53]. Additionally, no evidence of hepatic, renal, or cardiac toxicity emerged from this study and, perhaps surprisingly, the treatment cost to patients receiving creatine supplementation was recorded to be much lower than controls.

Studies using magnetic resonance spectroscopy have shown that loading protocols supplying 0.3 g/kg/day of oral creatine for 7 days elicit significant increases in brain creatine levels [88]. Moreover, creatine plasma concentrations and bioavailability may be enhanced by dietary choices, at least to some degree [89]. For example, evidence suggests that diets rich in animal-derived foods such as meat and fish, may provide up to 2 g of creatine per day [90,91]. As a result, individuals following a plant-predominant or plant-exclusive diet, such as vegetarians and vegans, may require higher doses of supplemental creatine to reach their recommended daily intakes if limited or no animal foods are consumed [92,93].

Oral supplementation represents a viable option to maintain stable creatine levels and/or effectively increase intracellular concentrations in various areas of the brain [94]. A recent double-blind, randomised cross-over trial in healthy young adults reported that a high single dose of creatine monohydrate (0.35 g/kg) was effective in attenuating changes in PCr/Pi (phosphocreatine/inorganic phosphate) and pH levels, while also reducing deficits in cognitive performance following partial sleep deprivation [95]. Importantly, these metabolic and cognitive benefits were observed starting at only 3.5 h from oral administration and lasted or were augmented up to 9 h after creatine ingestion. Positive effects of acute creatine supplementation have also been reported in several athletic populations, most notably in team and contact sports where frequent exposure to sub-concussive impacts has been suggested to deplete brain creatine stores [96]. For example, a single high dose of 0.05–0.1 g/kg of creatine has been shown to alleviate decrements in skill-based performance in sleep-deprived rugby players, and the magnitude of this positive result was found to be comparable to consuming caffeine doses of 1 or 5 mg/kg [54]. Similarly, creatine loading doses of 20 g/day for 7 days have been shown to improve cognitive function in experienced mountain bikers [55].

Based on these results, oral doses of approximately 20 g (~0.3 g/kg/day), distributed across the day to minimise saturation of uptake at higher doses (e.g., four 5 g doses, see Table 1) [97,98], likely represents the optimal supplementation strategy to maximise creatine absorption and availability in the brain pre- and post-TBI [99].

#### 4.1.2. Omega-3 Fatty Acids: DHA and EPA

Docosahexaenoic acid (DHA) and eicosapentaenoic (EPA) are long-chain omega-3 polyunsaturated fatty acids largely derived from seafood and marine algae. DHA preferentially accumulates in the brain and is critical to neurological function, injury risk, and prevention of neurodegeneration [100]. EPA, on the other hand, is mainly involved in vascular function, supporting overall cognitive health by decreasing systemic inflammation and improving oxygen and nutrient supply to the brain [50]. Accordingly, the potential benefits of omega-3 fatty acids for brain health may be in part explained by the increase in cerebral perfusion [50], while metabolites of DHA and EPA have been shown to modulate the inflammatory response after brain injury, driving resolution and repair [101].

The clinical utility of both DHA and EPA is largely attributable to their potent action against inflammation and neural damage induced by ROS. DHA and EPA supplementation has been shown to impede the synthesis of inflammatory cytokines such as interleukin-1 (IL-1) and tumour necrosis factor (TNF) [102]. Furthermore, it is believed that DHA and EPA may interfere with the expression of nuclear factor kappa b (NFKB), thus decreasing the transcriptional activation of several inflammatory genes [103]. To date, formal enquiries on the effects of DHA and EPA in the prevention and treatment of TBI are scant. Nonetheless, compelling research has recently emerged in relation to other neuropathologies, and can, to some extent, be generalised to acute traumatic head injuries [104,105,106].

From a structural standpoint, a number of human studies have examined the effects of DHA and EPA on brain and hippocampal volumes [48]. While cerebral atrophy tends to be quite widespread following TBI, recent observations have pointed to focal hippocampal atrophy as one of the most prominent outcomes [107]. Interestingly, not only have higher omega-3 indices been found to be positively associated with brain and hippocampal volumes, as demonstrated by compelling data from the Women’s Health Initiative [48], but significant increases in these structural measures across the lifespan have also been observed following DHA and EPA supplementation up to 2.2 g per day [108].

Patan, et al. [51] assessed the efficacy of different omega-3 supplementation protocols in supporting memory performance and consolidation in healthy adults and observed greater memory accuracy and recall speed with a combination of high dose EPA (900 mg) and moderate dose DHA (360 mg) when compared to both placebo and higher doses of DHA than EPA (900 mg and 270 mg, respectively). A separate study by Howe, et al. [50] in a cohort of hypertensive adults further evaluated the effects of 1600 mg of DHA and 400 mg of EPA over a 20-week period on a comprehensive battery of cognitive tests and found a positive association between cerebrovascular responsiveness, cognitive performance, and erythrocyte concentrations of EPA, but not DHA. Taken together, these studies indicate that EPA and DHA may be instrumental in facilitating different sets of neural functions and that EPA may be more strongly implicated in the promotion of positive cognitive outcomes. By contrast, DHA may play a critical role in regulating certain biomarkers of axonal injury following head trauma, particularly neurofilament light (NFL) [109]. In support of this hypothesis, Oliver, et al. [49] demonstrated that supplementing with two grams of oral DHA per day was sufficient to attenuate increases in serum levels of NFL in college football players, an athlete population known to be particularly prone to sub-concussive impacts [110,111].

Erythrocyte concentrations of DHA have been found to be reliable predictors of Alzheimer’s disease risk. For example, using data from the Framingham Offspring Cohort, Sala-Vila, et al. [52] reported a 49% lower risk of AD in the highest versus lowest quintile of DHA concentration in red blood cells. Importantly, this prospective observational study provides compelling evidence in favour of omega-3 supplementation as a potentially effective strategy to offset higher dementia risks, such as those attributable to genetics (e.g., APOE-ε4 carriership), or neurodegenerative cascades triggered by brain trauma [112]. Moreover, given that TBI-induced neurovascular injuries seem to accelerate some of the pathological mechanisms associated with Alzheimer’s disease (e.g., amyloid β production and perivascular accumulation) [113], such omega-3-based prophylactic approaches may be particularly beneficial for individuals at higher risk of head traumas.

Finally, another potential explanation for the neuroprotection provided by supplementing with omega-3 is its influence on arterial pliability. While the primary insult of a head injury is structural (e.g., tissue/axonal shearing), the cascade of secondary insults involves a mismatch between cerebral blood flow and metabolic demand, which are greatly affected by changes in the compliance and stiffness of cerebral vessels. Since the walls of downstream vessels have no external elastic membrane, the cerebral capillary network becomes vulnerable to intracranial and intravascular pressure surges. To protect against cell damage and death and provide adequate capillary bed perfusion, increased cerebral arterial compliance is required, and EPA (36%) and DHA (27%) supplementation has been found to assist within dyslipidemic adult populations aged 40–69, [114].

Despite the increasing recognition of DHA and EPA’s neuroprotective effects, the Standard American Diet (SAD) tends to be relatively deficient in these particular nutrients, with an estimated average intake of only 100 mg per day [115]. To address this nutritional inadequacy, oral supplementation is often recommended. Carefully designed supplementation protocols can effectively increase plasma and tissue levels of omega-3 fatty acids in a dose-dependent fashion up to approximately two grams, at which point saturation is commonly achieved [116]. The regular consumption of such doses of DHA and EPA does not seem to yield any major adverse effects [117]. Nonetheless, determining omega-3 status prior to supplementation is strongly encouraged to help identify the precise dose most suitable to address individual needs within the ranges typically recommended. Individuals at high risk of head traumas may therefore benefit from supplementing with 2 to 4 g of omega-3s per day (of which, ideally, 2 g from DHA) both as a preventative measure and as part of a comprehensive treatment protocol post-TBI.

#### 4.1.3. BCAAs

Branched-chain amino acids (BCAAs) are the essential amino acids leucine, isoleucine, and valine. BCAAs play an important role in brain metabolism by acting as nitrogen donors in a number of biochemical reactions, most notably the production of glutamate and GABA, two neurotransmitters strongly implicated in the physiopathology of TBI [118]. BCAAs are also known to interfere with the transport of the aromatic amino acids tryptophan and tyrosine across the blood–brain barrier via the large neutral amino acid transporter [119]. As a result, when blood concentrations of BCAAs rapidly increase, as happens following a protein-rich meal or oral supplementation [119], tryptophan and tyrosine uptake in the brain simultaneously decreases. Tryptophan and tyrosine act as precursors to other key neurotransmitters such as serotonin (and subsequently melatonin), and catecholamines [120]. Therefore, active competition for amino acid transporters directly affects the synthesis of multiple neurotransmitters, which may therefore alter functional changes in the brain [121].

To date, a number of animal models, particularly in mice, have shown that brain levels of BCAAs are markedly reduced following TBI [122], with significant improvements in TBI symptoms including reversal of cognitive deficits and sleep–wake abnormalities observed following BCAA supplementation [123]. These positive outcomes are thought to result from the BCAA-mediated restoration of network excitability through the correction of injury-induced imbalances in the release of both GABA and glutamate [124].

In humans, evidence suggests that the severity of typical clinical features of TBI correlates with the degree of BCAA suppression in the brain [125]. Indeed, levels of BCAA appear to be decreased in mTBI patients compared to healthy controls [126]. The beneficial effects of BCAA supplementation following head trauma have been extensively demonstrated in both mild [59] and severe TBI [56,57]. Importantly, the HIT HEADS trial by Corwin, et al. [59], which compared multiple BCAA doses to a placebo group, suggests that doses of BCAAs up to 54 g per day are associated with decreased concussion symptoms and greater return to baseline activity. These benefits showed a dose–response mechanism based on total calculated BCAA intake, though it must be acknowledged that the study drop-out rate was high across all treatment arms and based on the analyses presented it was not possible to determine a minimum effective dose. However, the study reported minimal adverse effects, reinforcing the idea that BCAAs may represent a safe and effective treatment option for TBI [127].

One potential consideration regarding high-dose BCAA supplementation is whether an imbalance in the uptake of tryptophan might impact melatonin production and therefore sleep, which is critical to recovery after TBI. This is one reason why carbohydrates are thought to be sleep-promoting under specific circumstances: the post-prandial insulin response decreases the availability of BCAAs and amino acid competition for uptake into the brain, increasing brain tryptophan and subsequent melatonin production. Conversely, elevation of BCAAs in healthy individuals is associated with sleep disturbances [128]. However, evidence to date does not support this concern in the setting of TBI. For instance, Corwin, et al. [59] described a trend towards improved sleep parameters in the BCAA groups. Similarly, Elliott, et al. [58] conducted a randomised controlled trial to evaluate the use of BCAAs on sleep in veterans with chronic TBI symptoms, with the intervention occurring years after the initial injury in some cases. The administration of 30 g of BCAAs twice a day was found to yield statistically significant improvements in subjective insomnia symptoms, as well as shorter sleep latency. These results are compelling given the ubiquity and severity of sleep disturbances post-concussion [129], and suggest that BCAAs may support sleep after TBI.

#### 4.1.4. Riboflavin and Other B Vitamins

Riboflavin, also known as vitamin B2, is one of the key coenzymes involved in the production of ATP. In light of the significant metabolic disruptions following TBI, the potential benefits of riboflavin and other B vitamins have therefore been investigated in models of TBI [130]. Studies in cortical controlled impact (CCI) rat models have shown that riboflavin supplementation improves behavioural function and decreases the concentration of glial fibrillary acidic protein, a marker of astrocytosis that occurs in both acute brain injury and neurodegenerative diseases [131,132]. The current literature supporting riboflavin supplementation for cognitive function in humans is sparse, however, clinical interventions employing a riboflavin protocol of up to 400 mg/day in the context of migraines have been reported as both safe and well-tolerated [133].

Kent, et al. [60] recently performed a randomised placebo-controlled trial evaluating riboflavin use in adults presenting with mTBI. Starting at 24 h after the injury, the intervention group received 400 mg of riboflavin per day for two weeks. Participants in the riboflavin group experienced a statistically significant reduction in the number of days to recovery compared to placebo (9.92 days vs. 22.2 days). However, it is worth mentioning that the above group difference was much smaller in the authors’ pilot study (10 vs. 13 recovery days in the treatment group vs. placebo), and that this discrepancy may be due to methodological factors such as sample size (the total number of participants, i.e., 60, was much lower than what was suggested by power calculations, i.e., 150), sport variation (not all sports were equally distributed in the study), and the fact that the original study design did not include records of initial symptom severity. If nothing else, this study at least demonstrated that riboflavin can be effectively combined with other supplements such as DHA and magnesium to better address the heterogeneity of post-concussion syndromes and related downstream effects, meaning that these nutrients can play complementary roles in various points of the cascade.

While other B vitamins have been extensively studied in animal models and clinical conditions that are somewhat adjacent to TBI, the available data in humans are limited. In rat models of brain ischemia, folate deficiency and resultant elevated homocysteine levels have been associated with increased oxidative stress [134]. Homocysteine is a circulating marker of one-carbon metabolism that is elevated when B vitamins (e.g., B2, B6, folate, and B12) are in insufficient supply [135,136]. Homocysteine promotes the hyperphosphorylation of tau, a well-described neuropathology hallmark of TBI [137], and is also a strong predictor of brain atrophy and cognitive decline with ageing. While not specifically tested in TBI populations, the benefit of homocysteine reduction via B vitamin supplementation in the context of age-related cognitive decline has been shown to be dependent on adequate omega-3 status [138,139,140], and it is possible that a similar interdependence of B-vitamin and omega-3 status is relevant to mitigation of TBI.

Supplementation with vitamin B12 has been shown to improve recovery and neurological function in a mouse model of CCI [141]. This positive result was thought to derive from the attenuation of endoplasmic reticulum stress-induced apoptosis, which is triggered by the accumulation of misfolded proteins in the endoplasmic reticulum and is a known contributor to the development of neurodegenerative conditions [142]. Recent findings have shown that vitamin B12 ameliorates endoplasmic reticulum stress and enhances nerve regeneration [143]. While compelling clinical evidence is needed to ascertain whether similar outcomes could be observed in human populations, given the notoriously safe profile of B vitamins and their pivotal role in facilitating essential metabolic functions in the body, insuring against inadequate intakes and deficiency is perhaps warranted in individuals at high risk for TBI.

#### 4.1.5. Choline

Citicoline, i.e., cytidine-5′-diphosphocholine or CDP-choline, is an intermediate compound in the synthesis of phosphatidylcholine, the major constituent of cell membranes in the brain [144]. Choline is also the precursor of the neurotransmitter acetylcholine. Accordingly, higher intakes of dietary choline have been consistently associated with significant decreases in several biomarkers of Alzheimer’s disease and an overall reduced risk of dementia [145,146,147]. Choline is commonly found in animal-based products such as meat, poultry, fish, eggs and dairy alongside cruciferous vegetables and beans. Nonetheless, data from the 2013–2014 National Health and Nutrition Examination Survey (NHANES) indicates that, in US adults, the average daily intake of choline is 402 mg in men and 278 mg in women, well below the established adequate intake (AI) of 550 mg and 425 mg, respectively [148].

Early experimental models in rats suggested that choline may act on critical targets of the inflammatory cascade following head traumas [149] and that chronic choline supplementation in the presence of ischemia may help preserve the integrity of the BBB by preventing phospholipid degradation in cell membranes [150,151]. Along the same lines, dose–response mechanisms have been observed, whereby 400 mg/kg of choline supplementation led to significant attenuations of brain oedema and BBB breakdown in rat models [152]. Whether similar benefits could be observed in human populations following this supplementation protocol remains unclear; however, if human-equivalent doses are derived based on body surface area, the resulting dose might be in the region of 70 mg/kg [153], which is 3–5 times higher than has been tested in humans so far, outlined below.

Animal studies have also examined the effects of relatively low dosages of exogenous choline and still reported potential benefits. For example, Dixon, et al. [154] found that as little as 100 mg/kg per day immediately after the experience of lateral controlled cortical impact yielded significant improvements in spatial memory performance in rats. As the authors suggested, this cognitive outcome may have been due to the appreciable increase in extracellular levels of acetylcholine in both the dorsal hippocampus and the neocortex resulting from choline supplementation [154]. More recently, Qian, et al. [155] evaluated the effects of 250 mg/kg of choline in rats 30 min and 4 h after closed head injury and, in agreement with previous studies, BBB breakdown and oedema were largely reduced. The supplementation protocol employed also appeared to attenuate axonal and myelin sheath dysfunction in the corpus callosum, while reducing hippocampal neuronal death 7 days from the time of injury [155].

When it comes to human research, findings regarding the effectiveness of choline in improving brain function are mixed [156]. Levin [61] conducted a double-blind randomised control trial looking at the role of choline in the management of post-concussion syndrome in mild TBI. In this study, the intervention group received 1 g of choline per day for one month and showed statistically significant increases in recognition memory scores compared to the control group. Interestingly, a separate study by Aniruddha, et al. [62] using the same supplementation protocol and an almost identical study design did not find significant differences between groups in common TBI symptoms (e.g., headaches, sleep disturbances, dizziness, and poor concentration). Nonetheless, the authors recognised that, for the majority of their participants, neurological damage was in fact minimal (GCS 15), meaning that the cholinergic pathways traditionally affected by the occurrence of TBI were likely not particularly compromised in this clinical cohort [62].

Notably, while two recent meta-analyses suggest that choline supplementation may improve cognitive rehabilitation and patient independence post TBI, irrespective of severity [156,157], the largest study on the subject conducted so far, the Citicoline Brain Injury Treatment Trial COBRIT, [63], found no evidence of improvements in either functional or cognitive status at 90 days post-injury when receiving 2 g of citicoline per day versus placebo. However, this double-blind randomised controlled trial presented several methodological limitations, the most striking being the inclusion of heterogeneous populations presenting with mild, moderate, and severe TBI, and whose precise pathophysiology and specific response to choline supplementation may have been significantly different. Additionally, the compliance rate with the treatment protocol was exceptionally low, with only 44.4% of patients having taken 75% of the daily dose [144].

Methodological issues aside, choline supplementation may be a preventative strategy of particular interest to athletic cohorts. For example, studies in college football players have shown that brain choline concentrations in the primary motor cortex diminish across the season in relation to head impact exposure, which in turn may contribute to increasing requirements or the accumulation of damage across multiple exposures [96,158].

In summary, the available scientific evidence suggests that chronic supplementation with choline is well tolerated and safe, with at least minor benefits being identified in both physiological and cognitive domains following head trauma [159]. As a result, given the durability and challenging treatment of some neurological and psychological symptoms associated with TBI, the utility of choline supplementation for rehabilitating populations seems warranted.

#### 4.1.6. Magnesium

Magnesium is an essential micronutrient known to facilitate several important metabolic processes in the human body, ranging from cell signalling and vascular function [160] to ATP production [161], protein synthesis [162], neuronal plasticity, learning and memory [163]. More broadly, magnesium deficiency is often associated with a host of comorbidities and health conditions including, but not limited to, type 2 diabetes, metabolic syndrome, elevated C-reactive protein, hypertension, migraines, headaches, and atherosclerotic heart disease [164]. Additionally, evidence suggests that magnesium may inhibit the activity of glutamatergic NMDA receptors, a direct target of commonly prescribed antidepressants [165] and a strong contributor to brain excitotoxicity after injury [27].

In the context of TBI, converging evidence from animal models suggests that magnesium is actively involved in many pathophysiological mechanisms triggered by acute head traumas and its contribution may be especially important in modulating excitotoxicity and promoting functional and cognitive recovery [166,167].

Following TBI in animal models, there is a well-documented decrease in magnesium concentrations specifically in central neurons [168]. Compelling research in rats further indicates that the extent of magnesium depletion is strongly associated with both the severity of head injury [169] and the level of behavioural disturbances experienced [170]. As such, magnesium supplementation post-injury has been shown to yield favourable outcomes in rat models by attenuating regional cerebral oedema, increasing brain and blood concentrations, decreasing ionised calcium levels, and improving behavioural deficits, memory and neuromotor function [171,172]. As far as cognition is concerned, several models of cortical injury in rodents, including cortical contusion injury, diffuse axonal injury and fluid percussion brain injury, have demonstrated the benefits of magnesium therapy on the recovery of cognitive function, particularly spatial and working memory performance [173].

In human studies of acute TBI, a randomised controlled trial in adolescents by Standiford, et al. [65] reported a significant reduction in post-concussion severity scores at 48 h post-injury following 400 mg of magnesium supplementation twice a day. In contrast, an earlier study by Temkin, et al. [64] did not observe significant neuroprotective effects of continuous infusions of magnesium for 5 days in patients presenting with moderate or severe TBI, thus raising the question of whether the TBI severity and/or the specific modality of magnesium administration may facilitate (or impede) the absorption of the nutrient, at least to some degree. Despite these discrepancies, oral magnesium supplementation after concussion at least appears to be safe and potentially effective in ameliorating typical symptoms and behavioural hallmarks of TBI. Importantly, forms of magnesium that supposedly have better penetration into the CNS, such as magnesium L-Threonate [174], have minimal evidence to support their use in this context above any other magnesium salt with good bioavailability (e.g., magnesium bisglycinate, sulphate, taurate, and malate).

#### 4.1.7. Blueberry Anthocyanins

Anthocyanins, from the Greek *anthos*, meaning “flower”, and *kyanos*, meaning “blue”, are water-soluble phytochemical compounds belonging to the flavonoid family and known for their characteristic polyphenolic pigments [175]. To date, more than 700 structurally different anthocyanin molecules have been identified, and their functions and health-promoting effects have been demonstrated in a number of conditions such as cardiovascular disease, metabolic syndrome, type 2 diabetes, and various types of cancers, but also vision and skin problems, inflammation, and neurodegenerative disorders [176,177]. Furthermore, anthocyanin extracts, specifically from Vaccinium myrtillus, have recently been shown to alleviate secondary brain injury in rats via anti-oxidative and anti-apoptotic mechanisms [178].

The behavioural deficits observed following TBI are thought to be at least partly caused by reductions in the release of brain-derived neurotrophic factor (BDNF) [179]. This mechanistic explanation is corroborated by animal studies showing that blueberry supplementation (5% *w*/*w*) post-injury helps maintain BDNF concentrations within normal ranges after experimental TBI, and that rising levels of BDNF are inversely correlated with maze time latency, thus supporting the idea that BDNF plays a pivotal role in the promotion of memory performance and cognition [180]. In the same study, blueberry supplementation appeared to protect against oxidative stress by reducing the concentration of 4-hydroxynonenal (4-HNE), the major end product of lipid peroxidation that is known to contribute to neural tissue damage, cell membrane dysfunction [181], and visuospatial memory deficits [182]. More specifically, the authors looked at alterations in levels of 4-HNE pre- and post-treatment and showed that 4-HNE levels increased in affected mice following injury, but subsequent blueberry supplementation effectively reversed this mechanism.

The efficacy of blueberry supplementation in improving cognitive function in humans has been evaluated in several randomised controlled trials, including developmental, healthy ageing and clinical populations [66,68,70,72]. The overall evidence from these studies suggests that supplementation can lead to improvements in attention, memory, and executive functions [71,183], although beneficial outcomes appear to be strongly dependent on the specific dosage administered to the study cohorts and/or the modality of delivery. For example, a double-blind cross-over study in children by Whyte, et al. [67] where participants consumed blueberry drinks containing 15 or 30 g freeze-dried wild blueberry powder or placebo reported significant improvements in cognitive performance in a dose–response fashion. Furthermore, a randomised double-blind, placebo-controlled trial in healthy older adults from the same research group indicated that 100 mg of wild blueberry purified extract facilitated better episodic memory performance and reduced cardiovascular risk factors over 6 months when compared to whole wild blueberry powder at both 500 mg and 1000 mg [69].

To date, no RCTs exist employing blueberries or anthocyanin extracts in patients after TBI. However, mechanistic support from preclinical research is promising, and blueberry anthocyanins have been shown to support cognitive function across a range of other human clinical studies. Most importantly, the doses employed are easily achievable from the consumption of anthocyanin-rich blueberries. As a result, supplementation could be considered exceptionally low risk if no allergy is present. From a practical standpoint, given the dose-dependent mechanisms illustrated above, supplementation with wild whole blueberries seems to be the best approach to reap the anti-inflammatory and antioxidant benefits of anthocyanin compounds. Indeed, wild-harvested low-bush blueberries (e.g., *Vaccinium myrtillus*, *Vaccinium angustifolium* or *Vaccinium myrtilloides*) have been shown to contain significantly larger amounts of polyphenols [184] and to be 3 to 5 times more effective against ROS, compared to commercially grown and harvested varieties, (e.g., *Vaccinium corymbosum*) [185]. Future enquiries exploring the potential benefits of blueberries in alleviating some of the symptoms related to TBI are encouraged. Further investigations should also consider other anthocyanin-rich fruits, such as strawberries and cranberries, which have been shown to be beneficial in the context of cognitive ageing [186,187]. In the meantime, a dietary intake of one cup (~150 g) of wild-grown blueberries per day could be recommended in both healthy individuals and populations at increased risk of TBI and/or dementia.

### 4.2. Non-Nutritive Compounds with Clinical Evidence in Humans in the Context of TBI

#### 4.2.1. Boswellia Serrata

*Boswellia serrata*, also known as Indian frankincense, is a resin-producing tree of the family *Burseraceae* (Genus Boswellia) native to India, North Africa, and the Middle East [175]. The biological compounds found in *Boswellia*, commonly referred to as Boswellic acids, have been shown to possess anti-inflammatory and neuroprotective properties across a range of preclinical studies [176], which may have important therapeutic implications both in neurodegenerative disorders [177,178,179] and for the treatment of TBI [180,181,182].

The mechanisms by which these biochemical agents may help reduce neuroinflammation post-TBI are believed to occur through the inhibition of NKFB. NFKB is a transcription factor that plays a central role in the expression of various inflammatory genes [183], alongside the activation and differentiation of both innate immune cells and inflammatory T cells [184,185].

With respect to human research, Meshkat, et al. [75] conducted a randomised control trial looking at the effects of 400 mg of boswellic acids in capsule form three times a day on individuals suffering from TBI of different severities, ranging from mild to severe, in the 3 months to 3 years prior to enrolment. Results showed significant improvements across a comprehensive battery of neuropsychological tests of cognitive function, including verbal and visual memory, attention, processing speed and executive function. Similarly, a pilot trial by Moein, et al. [73] on patients affected by diffuse axonal injury (a subset of TBI seen with blunt trauma to the head) [186] reported a near-significant positive trend in cognitive outcomes in the group randomised to three 360 mg capsules of boswellic acids per day compared to placebo, for whom spontaneous recovery during the study period did not approach statistical significance. Finally, a randomised, double-blind, placebo-controlled, pilot trial in ischemic stroke patients found that treatment with two 400 mg capsules of boswellic acids three times per day (i.e., 2400 mg/day) yielded significant recovery in neurological function during the 1-month follow-up, compared to placebo [74]. Additionally, at the 7-day mark post intervention the treatment group showed substantial decreases in inflammatory markers such as TNF-α, leukins IL-1β, IL-6, IL-8, and prostaglandin E2 (PGE2), thus showing that boswellic acids may be beneficial in improving clinical outcomes in the early phases of TBI recovery.

In conclusion, oral treatment with Boswellia serrata appears to be a promising intervention in patients presenting with TBI. Based on the current evidence, three doses of 400 mg per day in capsule form over a period of at least three months may be suitable to reduce neuroinflammation and improve several measures of cognitive health following head trauma.

#### 4.2.2. Enzogenol

Enzogenol is a natural pine bark extract sourced from New Zealand pine trees reported to have both antioxidant and anti-inflammatory properties [187]. To our knowledge, only two randomised control trials so far have investigated the use of enzogenol in TBI patients. In a small study by Theadom, et al. [76], supplementation was found to be both safe and well tolerated at dosages of 1 g/day for 6 weeks. Patients in the treatment group were found to have significantly fewer self-reported cognitive failures compared to the control group, and beneficial effects on this measure were sustained for 4 weeks after the end of the treatment (Table 2). Similarly, Walter, et al. [77] demonstrated that 1 g/day of enzogenol for 6 weeks reduces mental fatigue and sleep disturbances in young athletes with a history of sports-related concussions (six months to three years post-injury). Importantly, the authors used EEG recordings to monitor participants’ theta frequency bands in the frontal midline and parietal areas, which have been suggested to index cognitive resource allocation and mental fatigue, respectively, in several neuropsychological tasks [188,189]. Results indicated that, after six weeks of supplementation and compared to baseline, the treatment group experienced significant increases in frontal midline theta on the one hand, and smaller increases in posterior theta power on the other, thus suggesting that Enzogenol may improve neurocognitive function and physical symptoms in the chronic phase of concussive injury by acting on these specific physiological substrates [77].

#### 4.2.3. NAC

N-acetylcysteine (NAC), a derivative of the amino acid L-cysteine, is a pharmaceutical agent with the ability to increase intracellular concentrations of glutathione the body’s primary endogenous antioxidant [187]. Furthermore, NAC is thought to exert a protective effect against inflammation by suppressing the activity of NFKB, ultimately reducing levels of TNF-α and several interleukins [190]. With respect to TBI, several preclinical studies in rats demonstrated that NAC administration reduces markers of oxidative stress, ameliorates blast-induced changes in BBB integrity, and improves cognitive outcomes following controlled cortical impact or blast injury [191,192,193]. Importantly, human research, specifically in children with severe TBI, is currently underway, meaning that clinical evidence to corroborate these results in human subjects will hopefully be available in the near future (e.g., the Pro-NAC Trial—ClinicalTrials.gov NCT01322009) [194]. In the meantime, a double-blind randomised controlled study by Hoffer, et al. [78] in a cohort of 81 US active duty service members (18–43 years of age, 1 female) found benefits of NAC supplementation in the setting of blast exposure to mild TBI, as demonstrated by improvements in key neuropsychological measures, a reduction in the number of TBI symptoms, and shorter recovery times compared to placebo. The experimental protocol comprised 7 days of NAC supplementation at 4 g/day in the first 4 days (split into two 2 g doses), followed by 3 g/day (split into two 1.5 g doses).

#### 4.2.4. Melatonin

Melatonin is an endogenously produced neurohormone associated with sleep onset and regulation of several aspects of the central circadian supplement. Melatonin is also available as a supplement (or prescription medication, depending on the country), and has been investigated in the setting of TBI-related sleep disturbances [195]. In addition, melatonin is being explored as a neuroprotective agent in its own right due to its anti-inflammatory and both direct and indirect antioxidant properties seen in in vitro and in vivo preclinical models of TBI [196]. Recent systematic reviews have identified two clinical trials of melatonin supplementation after TBI—both focused on sleep quality. Kemp, et al. [79] randomised seven individuals with post-TBI sleep disturbances to 5 mg melatonin or 25 mg amitriptyline in a blinded crossover study pilot study. No differences were seen in any sleep metric, though statistical power was low, and patients reported increased daytime alertness when in the melatonin phase. Grima, et al. [80] performed a placebo-controlled crossover trial of 2 mg melatonin in 33 individuals experiencing sleep disturbances after TBIs that ranged from mild to severe. Significant improvements in sleep quality, sleep efficiency, and mental health were seen during the melatonin phase, but daytime sleepiness did not change [80]. Therefore, while the melatonin content of over-the-counter melatonin supplements are notoriously variable [197,198], melatonin may be considered for sleep support in the post-TBI period.

### 4.3. Additional Nutrients and Compounds with Lacking or Insufficient Evidence

A number of additional compounds have been considered in the context of TBI, although clinical evidence demonstrating their efficacy and/or utility is currently lacking or insufficient [199]. While future enquiries are needed to characterise the precise contribution of these biochemical agents to the treatment of post-concussion syndromes, this section presents a brief overview of their potentially neuroprotective properties.

In recent years, a number of peptides have emerged as promising lead agents for the treatment of TBI given their ability to cross cell membranes, high specificity and affinity for target molecules, and low immunogenicity and toxicity [200]. A notable example in the context of moderate to severe TBI is cerebrolysin, a neuropeptide derived from porcine brain tissue that has been increasingly gaining popularity for its perceived nootropic effects [187] and its several properties akin to those of endogenous neurotrophic factors [201]. The use of cerebrolysin in acute TBI has been investigated by Chen, et al. [202] and was found to elicit improvements in cognitive function when compared to placebo. Furthermore, no adverse effects were reported and the general recommendation from this study was to initiate the treatment within 24 h of injury. Similarly, a separate randomised controlled trial in moderate and severe TBI reported a small-to-medium-sized effect in favour of 50 mL/day of cerebrolysin supplementation for 10 days on a multidimensional ensemble of functional and cognitive measures assessed at day 30 and day 90 in the recovery phase [203].

Semax is the synthetic analogue of the endogenous hormone adrenocorticotropin and has been examined as a potential cognitive-enhancing agent in a number of animal models. Most notably, Dolotov, et al. [204] found that semax administration significantly increases the expression of BDNF in the rat hippocampus, while Medvedeva, et al. [205] demonstrated favourable outcomes of treatment with semax in immune regulation following ischemic brain injury in the same species. However, to this day, there is no human data on the use of semax in TBI patients.

Pramiracetam is a synthetic compound derived from piracetam, a pharmacological drug originally developed to treat motion sickness, but mostly known for its cognitive-enhancing effects [206]. While structurally similar to its precursor, pramiracetam is thought to be up to 30 times more potent in treating post-concussion symptoms such as amnesia and memory loss [207,208]. It is generally believed that the nootropic compounds belonging to the racetam family affect brain function and metabolism by modulating the excitatory and/or inhibitory processes of neurotransmitters and post-synaptic signals [208]. Pramiracetam in particular has been suggested to accelerate the cholinergic neuronal impulse flow in the septal hippocampal region, which may explain the enhancement in cognitive processes typically observed upon supplementation [209]. Notably, pramiracetam stands out from the other pyrrolidone acetamide nootropics for being a basic, rather than a neutral compound and for its rapid absorption and remarkable bioavailability [206]. Relatively high doses of pramiracetam up to 4 g per day have yielded mixed results specifically in Alzheimer’s disease [210,211]. However, an early, very small study in severe TBI reported that pramiracetam supplementation at intakes of 400 mg three times a day was well-tolerated and yielded significant improvements in recall memory performance that were maintained during an 18-month open trial period on the medication as well as during a 1-month follow-up period after discontinuation [212]. Despite these promising results, further research in larger TBI populations of different severities is required.

Curcumin is a diarylheptanoid polyphenol and the primary curcuminoid found in turmeric, with a history of uses across traditional medicine [213]. Curcumin has a range of putative neuroprotective effects including being antioxidant, anti-inflammatory, and neurodegenerative [214]. As a result, curcumin is included in a number of commercially available supplements and suggested protocols to support recovery after TBI including concussion. One particular issue is the bioavailability of curcumin, which is notoriously low, motivating the development of novel formulations of curcumin including nanoparticles and hydrogels by ourselves and others for the treatment of acute brain injury [215,216]. Despite potential issues with delivery to the brain, curcumin remains promising for TBI. A recent systematic review and meta-analysis of preclinical studies found several significant benefits of curcumin in rodent models of TBI [214]. Several human interventional studies of curcumin are also relevant, with multiple studies in older adults suggesting that curcumin may enhance—or at least slow the decline of—cognitive function, with related changes in pathology on brain imaging [217,218,219]. However, to our knowledge, no clinical studies in humans using curcumin have been performed, and none are currently registered with ClinicalTrials.gov as of the time of writing.

Exogenous ketones are a wide group of esters, salts, or fatty acids (specifically C-6 and C-8 medium-chain triglycerides) that increase circulating levels of the ketone bodies acetoacetate, β-hydroxybutyrate and acetone, which are naturally produced by the body in response to carbohydrate restriction and fasting [220]. During periods of acute brain injury, cerebral uptake of circulating ketones is increased, and these compounds can contribute approximately 70% of the brain’s fuel to make ATP [221]. Furthermore, experimental models suggest that ketone bodies may help improve recovery following TBI by addressing post-traumatic energy deficits, and by reducing inflammation and oxidative stress [222]. Nonetheless, human data are currently limited, and questions remain about appropriate dosing, duration, and timing of such treatment, whose accessibility may be substantially reduced by its relatively elevated costs [221]. As more and more evidence emerges to resolve these issues, in the years to come ketogenesis—by way of either ketogenic diets or exogenous ketone supplements—may come to assume a prominent role in the treatment of TBI.

Finally, preliminary evidence in relatively small human trials suggests that, following TBI, gluconeogenesis from lactate clearance is significantly increased by up to 71% compared to healthy controls [223], meaning that lactate mobilisation comes to assume a prominent role in the supply of energy to the body and its essential organs, including the brain, after trauma [224]. As a result, although large studies on exogenous lactate in TBI settings are currently not available, future investigations should ascertain whether arterial lactate supplementation may be employed to compensate for the depression in net glucose uptake commonly observed in the injured human brain [225].

Several other supplements were also considered due to social media interest and their inclusion in formulations purporting to support the brain before/after TBI but were not included in greater detail due to a lack of evidence related specifically to TBI within the goals of the manuscript. These included Lion’s Mane, Resveratrol, Acetyl L-Carnitine, Green Tea Extract, Coffee Fruit Extract, CoQ10, Nicotinamide Riboside, and other individual amino acids (L-Glutamine, L-Glycine).

## 5. Other Considerations—Caffeine and Sleep

While pre- and post-impact supplementation should be considered as important components of the armamentarium of mitigating the impact of concussions and more severe TBIs, two related (patho)physiological considerations should also be taken into account to minimise the impact of acute head trauma in the short term—caffeine and sleep.

### 5.1. Caffeine

Caffeine is a stimulant frequently in use amongst populations at high risk of TBI, namely athletes and military personnel, and may be exploited as a form of self-medication to combat post TBI fatigue and accelerate return to activity [226]. When considered through the lens of TBI, the most noteworthy effect of caffeine is the induction of neural vasoconstriction, leading to a substantial decrease in blood flow [227]. Eade [228] conducted a retrospective study to compare TBI symptoms in adolescents based on caffeine consumption. The results indicated that caffeine consumers were more susceptible to diminished emotional health, poorer sleep quality, and higher rates of depression and somatic symptoms compared to those who did not engage in this behaviour. As a result, while caffeine consumption—especially as tea and coffee and therefore confounded by the associated polyphenol intake—appears to be associated with a lower risk of neurodegenerative conditions at the population level [229,230,231], caffeine intake should be moderated in the symptomatic phase after TBI, as it may exacerbate some of its well-documented behavioural complications.

### 5.2. Sleep

Disruptions to normal sleep–wake cycles are often reported following TBI and represent a particularly challenging issue given the prominent role of adequate, restorative sleep in supporting immune function [232,233]. From a clinical standpoint, TBI patients experiencing sleep syndromes tend to exhibit slower recovery trajectories and more pervasive post-concussive symptomatology including headaches, chronic fatigue, and depression, alongside an array of cognitive deficits typically observed in several forms of neurodegeneration [234,235]. Studies using repetitive blast injury or closed head impact models of TBI, ranging from mild to severe, consistently showed decreased flow through the glymphatic vascular space, decreased expression and depolarisation of aquaporin-4 water channels on astrocytic end-feet, and compromised clearance of protein aggregates associated with TBI-related neurodegeneration such as amyloid beta and tau protein [236,237,238,239,240]. At present, human research examining the effects of TBI on the glymphatic system is still in its infancy and appropriate neuroimaging modalities to examine disruptions to cerebral flow during sleep need to be carefully validated [241,242].

Regardless of supplementation (e.g., with melatonin), common and relatively straightforward practices for effective sleep management strategies that should be considered in the post-TBI period include: adhering to a regular sleep schedule across the week, minimising bright light exposure in the evening hours, reducing the consumption of alcohol and caffeinated beverages, particularly in the second half of the day, and avoiding strenuous physical exercise, environmental distractors and brain stimulating activities, such as the use of social media and electronic devices, in the 1–2 h prior to bedtime. In conjunction with these measures, acupuncture, guided meditation, blue light exposure, and cognitive behavioural therapy (CBT) have also been shown to have beneficial effects on both the quantity and the quality of nocturnal sleep [243,244,245,246,247]. If these conservative methodologies fail to improve TBI-related symptoms, sleep medication may be considered under the supervision and expert guidance of health professionals. To date, research on the use of sleep-promoting agents in TBI populations is scant or inconclusive [248,249,250,251]. Nonetheless, the choice and supplementation protocol of prescribed sleep medication, especially the duration of the intervention and its relative dose, must be carefully evaluated against any potential adverse effects, ranging between mild drowsiness and fatigue to impaired psychomotor skills, substance dependence and/or abuse, as well as rebound insomnia and altered sleep architecture once the treatment is interrupted [252].

Given the wide-reaching consequences of sleep disturbances on an individual’s long-term health, particularly following TBI, sleep management must become a standard of care in the treatment of post-concussion syndromes and specific, evidence-based approaches are required to attenuate the negative repercussions of abnormal sleep patterns. Most importantly, in light of the strong interrelationship between sleep and immune function, targeted interventions aimed at correcting sleep-related impairments will likely promote the recovery of several other physiological, behavioural, and long-term neurocognitive effects of traumatic brain injury.

## 6. Future Directions—Synthesis, Context, and a “Left of Bang” Approach

In addition to improvements in safety, diagnosis, and treatment of those at high risk of and experiencing TBI, including concussions, and targeted supplementation strategies in line with current evidence, there are several other areas where additional work and research can further improve the outcomes of those with TBI. These include pre-morbid assessments in those at high risk of TBI, sometimes referred to as the period left of bang. Additional potential considerations include the management of physiology even in those with sports-related concussions not requiring hospitalisation. These areas of fertile ground are expanded upon and integrated below.

### 6.1. Premorbid Assessment of Nutrient and Metabolic Health Status

Of the supplements outlined above, several can be easily assessed clinically in those at high risk of TBI prior to impact. Though pre-impact supplementation has routinely only been performed in preclinical studies, with some exceptions such as DHA in American football players [49], there is strong reason to believe that improving nutrient status at baseline may mitigate the effects of later TBIs. Omega-3 status can be assessed using the omega-3 index, i.e., the percentage of EPA+DHA in the membrane of red blood cells. Professional and collegiate athletes are documented to have low omega-3 status on average, with population data suggesting improved cognitive outcomes at levels of 6–8%, or higher [48,100,253]. Despite most interventional studies using fixed dosages, greater benefit may be derived by tailoring omega-3 supplements based on individual nutrient status and diet, which show significant inter-individual variability.

Though true magnesium status can be difficult to determine, serum magnesium can be assessed easily, with studies suggesting that concentrations in the upper half of the normal range > ~0.9 mmol/L are associated with improved cognitive function in older adults [254,255]. Homocysteine can be assessed as a proxy for B vitamin status, as well as a risk factor in its own right. In the setting of cognitive decline, a level of at least <11 umol/L is associated with a lower risk of cognitive decline when omega-3 status is adequate [140,256,257,258,259]. Levels of folate, B12 (or the associated marker methylmalonic acid), riboflavin (or the associated marker erythrocyte glutathione reductase activity), and B6 (or the associated marker erythrocyte transaminase activity) can also be considered.

An additional critical consideration is vitamin D status. A retrospective study in TBI patients by Lee, et al. [260] showed significant improvements in cognitive measures upon vitamin D supplementation. Notably, the study patients were found to be vitamin D deficient at baseline. Given its high safety profile and ease of administration, vitamin D may therefore be considered as an additional supplement in high-risk individuals who may not reach their daily recommended intakes through sun exposure. This is particularly pertinent to those at high risk of TBI. For example, in a study of National Football League players, mean vitamin D status was insufficient (<30 ng/mL) [261,262] and lower vitamin D levels were associated with a higher incidence of fractures. Similar results have been reported in Division I collegiate football players. Testing of Vitamin D status, and supplementation, if necessary, is highly recommended.

Finally, the overall health of the individual should be considered. Blood glucose control after impact is an important predictor of outcome, especially in more severe TBI, as outlined in further detail below. Elevated blood glucose (e.g., prediabetes or diabetes) is also a component of the metabolic syndrome and may theoretically exacerbate a TBI-related injury if present at the time of impact. In high-level collegiate and professional athletes, metabolic syndrome components are common, particularly in positions and sports where being of higher body weight is advantageous such as heavyweight fight sports, American football linemen, etc. [263,264,265,266]. In the military, metabolic health tends to reflect the trends of the general population, where nearly 90% of adults have at least one component of the metabolic syndrome. While the effect of premorbid blood glucose control on recovery from concussions is not known, at minimum HbA1c could be assessed in those at risk of TBI as a measure of blood sugar control, with a target of <5.7% (normal).

### 6.2. Blood Glucose Regulation

Hyperglycaemia is traditionally associated with poor outcomes in TBI patients and has been shown to worsen neurological markers during decreased cerebral perfusion with hypoxia and/or ischemia [267,268,269]. The mechanisms by which hyperglycaemia exacerbates acute brain injury include increased oxidative stress, glutamatergic activity, and intracellular calcium levels, alongside the upregulation of both inflammatory and apoptotic pathways [270]. A recent meta-analysis by Zhu, et al. [271] looked at intensive blood glucose control, defined as blood glucose levels between 80 and 110 mg/dL, in the post TBI window. The authors examined seven RCTs on adults presenting with moderate and severe TBI and concluded that intensive blood glucose control was associated with improved neurological functions and shorter lengths of stay in an inpatient setting. While an increase in hypoglycaemic events was observed, the study points out that in most cases hypoglycaemia was not associated with detrimental outcomes. Along the same lines, a retrospective study by Fanchiang, et al. [272] evaluated the effects of blood glucose control in adults with TBI, ranging from mild to severe. As the authors point out, blood glucose levels at the time of admission below 110 mg/dL were associated with positive discharge outcomes, as indicated by lower scores on the Disability Rating Scale (DRS) compared to blood glucose concentrations above this threshold.

If not being managed in the hospital setting, the association between blood glucose control and TBI would suggest that minimising the consumption of highly processed carbohydrates with a high glycaemic index (either from refined foods or carbohydrate-based sports beverages) in favour of fibre-rich foods and high-quality protein in the period after head injury would be a reasonable approach that individuals should consider, both in the hospital setting as well as during convalescence at home after milder TBI. In this context, intermittent assessment of blood glucose using point-of-care devices to target maintenance of <110 mg/dl may also be reasonable. One caveat is that acute injury and illness itself drive hyperglycaemia as part of the stress and inflammatory responses [273]. Therefore, it may not be possible to completely prevent glycaemia excursions after injury, though best practice suggests that minimising nutritional or iatrogenic causes of large variations in blood glucose would be prudent. Future studies leveraging increasingly available technologies such as continuous glucose monitors in athletes after concussion may help to delineate the effects of endogenous and exogenous causes of hypoglycaemia on outcomes in this population.

### 6.3. Thermoregulation

Peri-injury thermoregulation is generally under-represented in the concussion literature but can affect many of the mechanisms of injury that one may hope to mitigate with a nutritional approach. Several decades of preclinical and clinical research have shown the critical role that thermoregulation plays in the outcome of acute brain injuries, with hyperthermia/fever thought to exacerbate inflammatory responses and the energy production deficit that occurs after injury [274,275]. For example, in a lateral fluid percussion model in rats, Sakurai, et al. [276] showed that both pre- and post-impact hyperthermia exacerbated the degree of injury.

Several large clinical trials have examined the therapeutic effects of active hypothermia after TBI, but overall, the results have been underwhelming except in the case of certain severe TBIs managed in the intensive care unit when hypothermia is used to manage intracranial pressure [277]. By comparison, the best evidence suggests that targeted temperature management to maintain normothermia while preventing fever is the most evidence-based approach to TBI [275,278,279], but this is often overlooked outside of the hospital setting. Commercial devices exist to cool the scalp after concussion and have been implemented within several professional sports including rugby and hockey, but all the studies suggesting benefits have been uncontrolled and unblinded, with no evidence that brain temperature is being decreased [280]. Alternative approaches that cool the neck may decrease brain temperature by reducing the temperature of the incoming carotid flow, but evidence favouring these devices is currently lacking.

Overall, it is likely that the prevention of hyperthermia is the most critical component of thermoregulation as it relates to outcomes after TBI, as appears to be the case for many acute brain injuries in adults [279]. As such, Tylenol or other anti-pyretic medications may be considered during the initial injury phase to minimise fevers. As TBIs often occur in heat-stressed environments like sports fields and exercise itself is associated with significant increases in core temperature, it has been suggested that exertion-related mild hyperthermia may itself be a risk factor for worse outcomes after a sport-related concussion [281]. However, little is known about the prevalence of fever after concussion, though one recent analysis of mTBIs in the military setting reported increased odds of reporting fever compared to non-injured controls [282]. From an interventional standpoint, while no trials of pre-exercise cooling have examined this question, strategies such as cold water immersion are already frequently used prior to and after heat-stressed athletic events as they can be ergogenic and decrease time to fatigue and well as accelerate recovery [283]. More research is certainly needed, but we remain optimistic that strategies already in place in the world of athletic performance may be able to be leveraged in the future to mitigate any effects of mild heat stress on incident concussion.

### 6.4. Integration and Synthesis

Based on the evidence outlined above, a workflow and integration of possible supplementation strategies with assessments of physiology and health before and after TBI is outlined in Figure 1. Overall, an approach based on positive asymmetry for mitigating the effects of TBI appears to lie in improving health prior to impact (e.g., “Left of Bang”) and focusing on compounds that have a well-known and low-risk safety profile as they are generally available in the diet. Further supplementation with other or isolated compounds can then be considered based on relevant individual symptomatology. Of note is the caveat that none of these approaches have been tested together as part of a comprehensive approach to TBI, and supplements have tended to be studied in isolation. The framework is therefore entirely theoretical and is proposed based on the available literature for high-risk scenarios where intervention is desired even though higher levels of evidence are not yet available.

## 7. Discussion

Our understanding of the pathophysiology of TBI has seen dramatic improvements in recent years. Accordingly, there is an increasing appreciation for the link between TBI and later neurodegenerative conditions and cognitive decline. These debilitating, often irreversible, conditions arguably constitute one of the major healthcare issues of the current generation [1,284] and, while public awareness continues to rise, concerted efforts must be made to reduce the risk and incidence of both TBIs of all degrees of severity and subsequent neurodegenerative disorders. Notwithstanding, the therapeutic interventions and pharmaceutical treatments available so far have largely been ineffective or, at best, underwhelming [22]. Compelling mechanistic and clinical data point to the field of nutraceuticals as a promising, scalable solution [24]. However, the use of supplements in the context of TBI warrants careful consideration. If supplementation is being considered, the starting point should be the augmentation of nutrients that are available from the diet, including testing (e.g., of vitamin D or homocysteine status) as necessary. This represents the lowest risk level of intervention with a high potential for benefit. Additional supplementation may be considered but should be selected based on the results observed in relevant clinical trials.

The purpose of this review was to summarise the scientific evidence on the preventative and/or recuperative effects of selected micronutrient and biochemical agents on the pathophysiological symptoms associated with TBI. Such critical analysis allowed us to identify a number of compounds of particular interest for individuals at high risk of head trauma, as well as promising areas of research deserving further investigation. Based on our assessment of the available literature, there appears to be no reason to propose different dietary interventions based on gender or age. However, in order to ensure the efficacy of both preventative and therapeutic strategies, people affected by—or at high risk for—brain injury should trial the supplementation protocol that best suits their individual needs given that a certain degree of inter-individual variability in the physiological response to each nutrient is to be expected.

Perhaps not surprisingly, the current body of literature points to biochemical compounds such as omega-3 fatty acids, magnesium, and anthocyanins as potential neuroprotective agents against the brain’s excitotoxic and inflammatory response post-TBI [48,51,285,286]. Similarly, evidence suggests that creatine monohydrate allows for maintaining adequate ATP levels to address the brain’s high energy demands post-TBI [83,86], while choline attenuates brain oedema and helps preserve the integrity of the blood–brain barrier and cellular membranes [152]. Several studies have further demonstrated that BCAA supplementation restores network excitability and corrects potential injury-related imbalances in the release of GABA and glutamate [124], whereas Boswellia serrata downregulates the production of inflammatory cytokines [287]. Enzogenol and NAC have been found to support cognitive function and decrease markers of oxidative stress and neuroinflammation following concussion [76,78]. Finally, though only one high-quality study is available currently [80] the use of melatonin in those with post-TBI sleep disturbances is worth considering. In summary, the numerous and remarkable benefits of these micronutrients are well-documented and understood, specifically in relation to brain health and cognition [108,288,289], and such evidence justifies their use as safe, potentially therapeutic interventions in TBI clinical settings.

By contrast, the limited number of human studies assessing the safety and efficacy of a few newly discovered and/or less investigated micronutrients (e.g., exogenous ketones, lactate, semax, cerebrolysin, pramiracetam) constitutes one of the greatest limitations of the present review. Emerging results from preclinical data and animal studies provide insight into the interplay between these biochemical compounds and some of the physiological processes triggered by brain injury, however their generalisability to human cohorts remains questionable [290,291]. Ethical concerns in the design and implementation of testing protocols on human subjects constitute a significant barrier to the advancement of clinical research [292,293], which is why meaningful changes in pharmacological approaches to TBI management have historically been quite slow and difficult to endorse.

We must acknowledge that the accumulation of sufficient clinical evidence to confirm causal associations between dietary interventions and TBI-specific health outcomes is held up by the common methodological limitations of nutrition studies. These include small sample sizes, significant variability in the dosages employed in each trial, and short observation periods, thus casting doubt on the long-term efficacy and safety of tested protocols. Issues related to the accuracy and reliability of assessment methods capturing nutrition data warrant careful consideration as well, not only when employing subjective tools (e.g., self-reports) due to potential recall, respondent and expectation biases, but also in the case of objective measurements such as biochemical assays if no standardised testing and collection procedures are in place. Missing, inaccurate or confounding information within the study data is problematic for researchers and clinicians alike and can have strong repercussions on the conclusions and potential treatments derived from experimental observations. Accordingly, well-validated tools to comprehensively assess participants’ health history and physiological response to targeted therapies (including neuropsychological and cognitive testing) are required to assist with the diagnosis and management of TBIs more broadly [294].

Crucially, any attempt to conduct a comprehensive assessment of nutritional and pharmacological agents for the prevention and treatment of TBI must take into account that the supplement industry is not appropriately regulated by the FDA. Such shortage of procedural rigor in the manufacturing, third-party testing, and subsequent distribution of dietary supplements is extremely problematic [295]. Although it may be argued that the overwhelming abundance of non-prescription medicines currently available has significantly increased accessibility to quick and effective treatment options for a number of ailments, including those experienced following TBI, the ensuing, and all too frequent, self-prescription of over-the-counter supplements inevitably exposes consumers to substantial health risks and biohazards [296,297].

The lack of clear-cut legislations to ensure that the production of dietary supplements meets specific quality and safety standards, and the absence of specifically appointed regulatory bodies to enforce them constitute a cause of major concern for clinicians and consumers alike [298,299]. In other words, there is currently little to no guarantee that the many dietary supplements commonly found on supermarket shelves actually contain the biochemical compounds stated on their label, in the correct quantities and/or concentrations, and are free from potential contaminants, and other harmful or toxic by-products [300,301,302].

Such inaccuracy at the production end of the supplement chain is non-trivial given that biochemical compounds generally considered safe, and health-promoting may soon become harmful when consumed in supraphysiological doses and have irreparable consequences on an individual’s health [303,304]. Moreover, high concentrations of certain metabolites that could be derived from prohibited substances deliberately or unknowingly added to a proprietary supplement may be a cause for concern in professional and recreational settings. For instance, this would be most relevant in athletic populations where mandatory drug testing is routinely conducted to ensure compliance with national and/or international sport regulations [305,306].

Consumers may also be hesitant to implement an extensive supplementation protocol for financial reasons [307] and considering the limited commercial availability and/or loose regulation of certain newly discovered biochemical agents at this point in time (e.g., peptides and ketones). As formal enquiries continue to be conducted and more scientific evidence becomes available, we shall at least hope for some of these nutrients to be formally included in TBI treatment and management protocols, and that the healthcare system may ultimately be expected to take charge of some of these costs.

As a rule of thumb, consumers should be encouraged to fulfil their micronutrient requirements by consuming whole foods rather than synthetic ingredients whenever possible and limit exogenous supplementation to bridge any remaining dietary gaps. This may minimise the need for active supplementation. For example, the optimal prescribed dose of EPA/DHA could be accomplished through the consumption of wild caught fatty fish 2–3 times per week [308], although the sustainability of this approach may be challenging for most individuals. Similarly, almonds, legumes, and dark leafy greens are well-known rich sources of magnesium [309], while just two eggs provide more than 50% of the recommended daily intake of choline [310]. Finally, it should be noted that most micronutrients tend to be more easily utilised by the body when consumed as part of their natural whole food matrix, as they often work in synergy with other vitamins, minerals, co-factors, and enzymes that may be present in the food and therefore concomitantly absorbed. For example, wild low-bush frozen blueberries appear to be a better source of anthocyanins than almost any commercially available product [311].

## 8. Conclusions

In summary, the scientific evidence currently available points to several nutritional compounds and dietary supplements as valuable biochemical and pharmacological aids in the prevention and treatment of traumatic brain injury. In light of the considerable physio-psychological and financial burden imposed by both TBI and cognitive decline on our society, concerted effort must be made to design and implement systematic, evidence-based approaches to reduce the incidence of these debilitating conditions and mitigate the effects of their wide-reaching health sequelae. If being considered at this stage, supplements should also be included in the context of a broader holistic approach that includes pre-impact health assessments (if possible), temperature, sleep, and blood sugar management, as well early return to physical activity [312]. While the critical analysis and suggested doses outlined in this review represent a first step in this direction, further clinical work is required to deepen our understanding of the pathophysiological mechanisms associated with TBI, explore promising avenues for its timely and effective management, and hereby inform policymakers in the development of appropriate prevention strategies.

## Figures and Tables

**Figure 1 nutrients-16-02430-f001:**
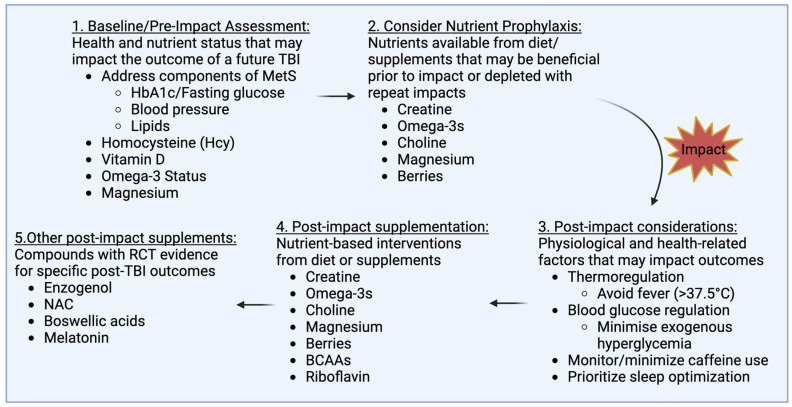
Proposed approach to assessment and intervention in those at high risk of TBI. (1) A full health assessment to screen for and address nutritional and metabolic health markers that may be associated with worse outcomes after TBI. (2) Consider a “left of bang” nutritional approach focused on compounds available in the diet that may mitigate the effects of impact or whose levels or requirements are potentially modified by the accumulation of multiple smaller impacts over time (e.g., creatine, choline, omega-3s). (3) Immediately post-impact, a focus on physiological parameters and sleep is likely to assist in recovery. Additionally, (4) and (5) show supplementation with higher doses of diet-derived compounds may be considered, in addition to other supplements with symptom-specific benefits such as melatonin. Figure made with Biorender.

**Table 2 nutrients-16-02430-t002:** Summary of the proposed nutritional and supplementation protocol for the prevention and treatment of TBI.

Nutrient/ Biological Compound	Recommended Intake and Supplementation Strategy	Adverse Effects	Food Sources and Corresponding Amounts Per 100 g
*Nutritive compounds derived or available from food*			
Omega-3 fatty acids (DHA and EPA)	2–4 g/day of combined DHA and EPA (of which 2 g from DHA)	None	Salmon (2.15 g/100 g cooked) Herring (2 g/100 g cooked) Sardines (1.4/100 g canned) Mackerel (1.2/100 g cooked) Trout (1 g/100 g cooked)
Creatine monohydrate	4 × 5 g/day (20 g/day total)	Potential mild GI distress with doses > 10 g	Beef (600 mg/100 g cooked) Chicken (520 mg/100 g cooked) Herring (1.1 g/100 g cooked) Salmon (600 mg/100 g cooked) Tuna (535 mg/100 g cooked) Cod (400 mg/100 g cooked)
BCAAs	Up to 54 g/day	Potential mild GI distress with supplemental daily doses > 45 g	Meat and poultry (3.6 g/100 g) Dairy products (2.37 g/100 g) Cereals and pasta (1.17 g/100 g)
Riboflavin	400 mg/day	None	Beef liver (3.4 mg/100 g) Fortified cereals (4 mg/100 g)
Choline (as CDP-choline/Citicoline)	1–2 g/day	None	Beef liver (419 mg/100 g) Hard boiled eggs (294 mg/2 eggs) Roasted soybeans (125 mg/100 g) Chicken breast (85 mg/100 g)
Magnesium (any bioavailable form)	400 mg/day	None	Pumpkin seeds (184 mg/100 g roasted) Chia seeds (131 mg/100 g) Almonds (94 mg/100 g roasted) Spinach (78 mg/100 g boiled)
Blueberry anthocyanins	250–400 mg/day	None	Low-bush wild blueberries (487 mg/100 g)
*Non-nutritive compounds*			
Boswellia serrata	3 × 400 mg/day	None	N/A
Enzogenol	1 g/day	None	N/A
NAC	4 g/day for 4 days (2 × 2 g), then 3 g/day (2 × 1.5 g).	None	N/A
Melatonin	2 mg at night	None	N/A

GI = gastrointestinal; N/A = not applicable.

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
