# Peer review of "Mitigating Traumatic Brain Injury: A Narrative Review of Supplementation and Dietary Protocols"

_nutrients, 2024, doi:10.3390/nu16152430_

Round 1
Reviewer 1 Report
Comments and Suggestions for Authors
This is a strong, well-written review of recent literature on the possible use of selected nutritional supplements for patients with TBI. Strengths of this manuscript include:
2. There is an appropriately brief discussion of the neurobiological mechanisms associated with TBI.
3. While supplying information about some relevant pre-clinical work, this manuscript is clearly focused on human TBI and the clinical work to evaluate the efficacy of nutritional supplements.
4. The manuscript clearly lays out recommendations for supplementation doses, but also provides food sources that could help meet the recommendations. This is very helpful to both clinicians, but also to basic researchers who could use this information to design additional pre-clinical studies. As a sidenote the recommendation on line 955 that the “optimal prescribed dose of EPA/DHA could be accomplished through the consumption of wild caught fatty fish 2-3 times per week”, is possible, but probably not realistic for most Americans.
5. The section of sleep seems somewhat misplaced. Consider moving this to the section on melatonin.
Author Response
1. This is a strong, well-written review of recent literature on the possible use of selected nutritional supplements for patients with TBI. Strengths of this manuscript include:
- There is an appropriately brief discussion of the neurobiological mechanisms associated with TBI.
- While supplying information about some relevant pre-clinical work, this manuscript is clearly focused on human TBI and the clinical work to evaluate the efficacy of nutritional supplements.
- The manuscript clearly lays out recommendations for supplementation doses, but also provides food sources that could help meet the recommendations. This is very helpful to both clinicians, but also to basic researchers who could use this information to design additional pre-clinical studies. As a sidenote the recommendation on line 955 that the “optimal prescribed dose of EPA/DHA could be accomplished through the consumption of wild caught fatty fish 2-3 times per week”, is possible, but probably not realistic for most Americans.
Response: We thank the reviewer for providing such positive feedback. With respect to their comment about EPA/DHA dietary recommendations on line 955, we have now elaborated on this statement as follows (lines 1122-1124):
“Optimal prescribed dose of EPA/DHA could be accomplished through the consumption of wild caught fatty fish 2-3 times per week, although the sustainability of this approach may be challenging for most individuals”
2. The section of sleep seems somewhat misplaced. Consider moving this to the section on melatonin.
Response: We thank the reviewer for raising this point. We do acknowledge that the idea of adequate sleep being critical for the treatment of TBI is briefly touched upon when discussing melatonin; however, the later section focused specifically on sleep is aimed at providing a broader, more comprehensive perspective on the pivotal role sleep plays in the management of TBI above and beyond dietary supplementation. In this section, we take the opportunity to outline important mechanisms of sleep related to TBI outcomes and point to several potential strategies and behavioural practices to improve both sleep quantity and quality in individuals experiencing sleep disturbances following brain injury, not all of which is directly relevant to the evidence for melatonin. Therefore, in order to keep the structure of the manuscript consistent when discussing individual supplements, we have not combined these sections.
Reviewer 2 Report
Comments and Suggestions for Authors
Authors present a topic of interest. Please find below some points for a major revision:
Please revise the abstract. A more structured and focused abstract is necessary. For example, focus more in the results.
The first two sentences in the Introduction section are vague.
The last paragraph of the Introduction (limitations should be moved to the Discussion).
A more detailed table with the main information about the supplements would be helpful.
Authors can also discuss the missing information in relevant research about the premorbid neuropsychological status of TBI patients and the methodological difficulties this creates for researchers and clinicians in their conclusions along with the lack of relevant tools (regarding cognitive, emotional, behavioral testing) for this population (for a relevant article on this, please use: doi: 10.1111/acem.13171).
A reconstruction of the discussion could also benefit the paper. Summary should be replaced by Conclusions.
Comments on the Quality of English LanguageMinor language editing
Author Response
1. Please revise the abstract. A more structured and focused abstract is necessary. For example, focus more on the results.
Response: We thank the reviewer for this suggestion. We have now modified our abstract to include more specific detail on some of the nutrients under investigation. We have also added more information regarding the additional considerations and assessments that should accompany nutritional interventions as part of a more effective and comprehensive management plan. We believe that these modifications present a clearer picture of what is being discussed in the main text by pointing to key areas of focus.
2. The first two sentences in the Introduction section are vague.
Response: These introductory sentences have now been rephrased as follows (lines 44-50):
“Traumatic brain injuries (TBIs) constitute a significant public health issue, and a major source of disability and death in the United States and worldwide [1]. While precise incidence rates can be difficult to quantify due to the elusive nature and frequent underreporting of many head traumas, recent estimates clearly point to upward trends in the prevalence of TBIs across the globe. For example, according to the Centers for Disease Control and Prevention (CDC), in 2014 there were approximately 2.87 million TBI-related emergency department visits, hospitalisations, and deaths in the US alone, representing a 53% increase from 2006 [2].”
3. The last paragraph of the Introduction (limitations should be moved to the Discussion).
Response: We thank the reviewer for the suggestion. In this short paragraph we simply wanted to point out that our review was going to be mainly focused on mild brain injuries, while also acknowledging that most of the scientific evidence relates to more severe head traumas. This is largely because sport related concussions, where the strategies discussed in the manuscript may have the greatest impact, tend to be very mild TBIs without loss of consciousness. Therefore we wished to acknowledge upfront that we would be including literature from more severe TBIs where the spectrum of injury will be different from milder concussions. We understand that this may cause some confusion, and we have therefore modified the last two paragraphs of the Introduction to clarify this point as follows (lines 112-134):
“The scope of the present narrative review is to summarize the current literature on the neuro-protective effects of supplements in the context of TBI and downstream neurocognitive and other related symptoms, and to provide an evidence-based overview of current supplementation and dietary protocols that may be considered in individuals affected by - or at high risk for - concussion and more severe head traumas. As such, emphasis will be largely placed on human studies. However, when human data on specific nutritional interventions are not available, we will discuss evidence from animal models to illustrate potential benefits and mechanisms of action for future studies to investigate.
While a number of investigational compounds may qualify as supplements relevant to TBI, we have chosen to focus on those that are readily available to the consumer. Importantly, though most of the relevant research has been conducted in TBIs of significant severity, we argue that clinical evidence favoring supplemental efficacy could be extended to milder brain injuries based on mechanistic plausibility, and because of the strong safety profile of the nutritional compounds under examination. In fact, the heterogeneous nature of the injury, especially within the category of mTBI, warrants careful consideration. “Mild” TBIs include everything from sport-related concussions where the athletes independently walk off the field to much more severe injuries that involve significant loss of consciousness (up to 30 mins) and the need for hospitalisation. The narrative approach of the present review is therefore aimed at broadening the scope as much as possible while searching for positive asymmetry – interventions with low risk where there is potential for benefit despite the absence of large injury- or outcome-specific randomised con-trolled trials. As emerging research suggests, supplementation and dietary approaches have the potential to mitigate the physical, neurological, and emotional damage inflicted by TBIs across the spectrum of severity.”
4. A more detailed table with the main information about the supplements would be helpful.
Response: We thank the reviewer for this suggestion. We included two tables in our paper to illustrate (i) the mechanisms of action by which nutrients of interest may have beneficial effects on TBI-related outcomes, alongside relevant research studies to indicate the level of strength supporting this clinical evidence (Table 1), and (ii) suggested supplementation protocols for the prevention/treatment of TBIs based on the findings of our critical analysis of the literature, including potential side effects and best dietary sources (Table 2). Taken together, we believe that these tables provide sufficient detail in relation to the nutrients and biochemical compounds under investigation. However, we would be happy to include additional information after re-review, should the reviewer have more specific information that they feel should be provided.
5. Authors can also discuss the missing information in relevant research about the premorbid neuropsychological status of TBI patients and the methodological difficulties this creates for researchers and clinicians in their conclusions along with the lack of relevant tools (regarding cognitive, emotional, behavioural testing) for this population (for a relevant article on this, please use: doi: 10.1111/acem.13171).
Response: We thank the reviewer for this suggestion. We have now integrated this information in our Discussion (lines 1084-1087).
6. A reconstruction of the discussion could also benefit the paper. Summary should be replaced by Conclusions.
Response: We thank the reviewer for this suggestion. We have now moved the Discussion after the section on Future directions and included a short paragraph pointing to methodological limitations of the available body of literature (lines 1073-1087). We believe that this new placement and addition effectively address the reviewer’s concerns. We have also changed the title of the last section from “Summary” to “Conclusion”.
Reviewer 3 Report
Comments and Suggestions for Authors
The manuscript is a narrative review of the current literature on supplementation and dietary protocols in traumatic brain injury. The scope of the manuscript is to summarize the existing literature on this topic. The manuscript is well-written and considers the current literature with a fair balance. The paper presents a balanced review of the existing literature helping clinicians to considered the best approach to help people after a traumatic brain injury. Overall I think the authors have done a very good job and the paper is very well structured. Point 6.4 and 7 represent relevant and significant conclusions of the manuscript, with suggestions for future steps. I have only a few points to underline to the authors for the revision of their manuscript:
- from a methodological point of view, the authors used a narrative approach, so no methodological improvement is possible. However, please include a statement at the end of the introduction about the approach you used for the selection of the literature
- I think Table 1 is misplaced in the wrong paragraph
- please include a paragraph on the limits of the current literature by the end of the paper
Author Response
The manuscript is a narrative review of the current literature on supplementation and dietary protocols in traumatic brain injury. The manuscript is well-written and considers the current literature with a fair balance. Overall, I think the authors have done a very good job and the paper is very well structured. I have only a few points to underline to the authors for the revision of their manuscript:
1. Please include a statement at the end of the introduction about the approach you used for the selection of the literature
Response: We thank the Reviewer for the suggestion. We have now included a short Methods section following the Introduction to briefly explain our literature search strategy (lines 135-145).
2. I think Table 1 is misplaced in the wrong paragraph
Response: We placed Table 1 at the end of Section 2 to provide a clear, comprehensive summary of the mechanisms of action by which our nutrients and compounds of interest are thought to exert their beneficial effects on TBI-related health outcomes. Additionally, by pointing out the types and characteristics of available clinical studies, we wanted to inform the reader on the strength of evidence supporting these purported physiological mechanisms, before discussing relevant results in more detail in the corresponding sections thereafter.
3. Please include a paragraph on the limits of the current literature by the end of the paper.
Response: Following the suggestions of both reviewer 2 and 3, we have now moved the Discussion after the Future Direction section, and we have included a short paragraph about methodological limitations of the current body of literature (lines 1073-1087).